# Emerging Therapeutic Landscape of Peripheral T-Cell Lymphomas Based on Advances in Biology: Current Status and Future Directions

**DOI:** 10.3390/cancers13225627

**Published:** 2021-11-10

**Authors:** Maliha Khan, Felipe Samaniego, Fredrick B. Hagemeister, Swaminathan P. Iyer

**Affiliations:** Department of Lymphoma/Myeloma, University of Texas, MD Anderson Cancer Center, Houston, TX 77030, USA; fsamaniego@mdanderson.org (F.S.); fhagemei@mdanderson.org (F.B.H.)

**Keywords:** PTCL, non-Hodgkin’s lymphoma, novel therapies, targeted therapies, recent advances, current treatment

## Abstract

**Simple Summary:**

Peripheral T-cell lymphoma is a rare but aggressive tumor. Due to its rarity, the disease has not been completely understood. In our review, we look at this lymphoma at the molecular level based on available literature. We highlight the mechanism behind the progression and resistance of this tumor. In doing so, we bring forth possible mechanism that could be exploited through novel chemotherapy drugs. In addition, we also look at the current available drugs used in treating this disease, as well as highlight other new drugs, describing their potential in treating this lymphoma. We comprehensively have collected and present the available biology behind peripheral T-cell lymphoma and discuss the available treatment options.

**Abstract:**

T-cell lymphomas are a relatively rare group of malignancies with a diverse range of pathologic features and clinical behaviors. Recent molecular studies have revealed a wide array of different mechanisms that drive the development of these malignancies and may be associated with resistance to therapies. Although widely accepted chemotherapeutic agents and combinations, including stem cell transplantation, obtain responses as initial therapy for these diseases, most patients will develop a relapse, and the median survival is only 5 years. Most patients with relapsed disease succumb within 2 to 3 years. Since 2006, the USFDA has approved five medications for treatment of these diseases, and only anti-CD30-therapy has made a change in these statistics. Clearly, newer agents are needed for treatment of these disorders, and investigators have proposed studies that evaluate agents that target these malignancies and the microenvironment depending upon the molecular mechanisms thought to underlie their pathogenesis. In this review, we discuss the currently known molecular mechanisms driving the development and persistence of these cancers and discuss novel targets for therapy of these diseases and agents that may improve outcomes for these patients.

## 1. Introduction

Peripheral T-cell lymphomas (PTCLs) are a diverse set of aggressive T-cell lymphomas that arise from mature T cells [1]. Most PTCLs are associated with a poor prognosis [2]. PTCLs constitute 15–20% of aggressive non-Hodgkin’s lymphomas (NHLs) and 5–10% of all NHLs [3]. PTCL has a diverse morphology, and definitive markers of PTCL subtypes are scarce, making the diagnosis and classification of PTCL complex [4]. The 2016 World Health Organization classification system describes 27 different types of PTCL [5].

Most PTCLs are treated similarly due to the lack of specific targeted therapies for different PTCL subtypes. This, in turn, is due to an inadequate understanding of the pathobiology of these tumors. However, recent advances in whole-genome sequencing and gene expression profiling (GEP) have enabled us to better elucidate the pathogenesis of PTCL and differentiate among its various subtypes [6]. Recent studies have uncovered individual molecular signatures that may not only provide prognostic information about the disease but may also provide potential therapeutic targets. Multiple promising therapies are being tested and based on evidence from clinical trials; these therapies are being incorporated into the currently recommended treatment regimens.

In this review, we highlight the underlying pathobiology of PTCL, an improved understanding of which would potentially uncover novel therapeutic targets. We also describe the current treatment standards for PTCL and focus on novel targeted therapies that have demonstrated efficacy in recent clinical trials.

## 2. Origins of PTCL

Understanding the origin and development of T-cell lymphoma requires the knowledge of normal T-cell development and function. Mature T cells can be divided into two main subtypes based on their type of T-cell receptor (TCR) expression. Most T cells are αβ T cells, which express the αβ TCR chains, whereas fewer than 2% of T cells are γδ T cells, which express the γδ TCR chains [7]. T cells are further differentiated into subtypes through a process controlled by transcriptional regulators and cytokines, which exhibits significant plasticity under environmental pressure. The interaction of a normal T cell with an antigen-presenting cell (APC) forms a T-cell APC immunological synapse that employs multiple signaling pathways to induce T-cell activation [8]. Most notable among these pathways are the JAK-STAT, PI3 K-AKT, and MAPK signal transduction cascades [9]. These pathways are boosted by the activity of costimulatory molecules such as CD28 [7]. Epigenetic mechanisms also play a role in the immune response, increasing its complexity [9]. Dysfunction in these pathways could be at the core of lymphomagenesis, and hence, these pathways are being investigated with genomic and molecular approaches.

PTCLs can acquire unique phenotypes owing to mutations that modulate the transcription or the cytokine-driven plasticity of T cells and other tumor microenvironment signals. This modulation may lead to features that are dissimilar to those of the tumor’s cell of origin [10]. Hence, researchers have not yet established the precise cell of origin for different types of PTCL. However, global GEP, which has been used to define some types of B-cell lymphomas, has also been used to define some types of PTCLs [11]. For example, GEP has been used to define angioimmunoblastic T-cell lymphoma (AITL), a PTCL that has a GEP similar to that of T-follicular helper (Tfh) cells, which provides clues regarding its cell of origin [12]. Genetic signatures have been used to identify subgroups of PTCL-not otherwise specified (PTCL-NOS), a group of unclassified PTCLs [13]. The expression of multiple transcripts that match the genetic signature of normal natural killer (NK) cells has also been identified in some PTCLs [14].

## 3. PTCL Subtypes and Their Pathogenesis

This section highlights the biological underpinnings that characterize different PTCL subtypes. Common pathogenic mechanisms associated with PTCL subtypes are listed in Table 1. Some of these mechanisms are also highlighted in Figure 1.

### 3.1. AITLs and PTCLs with the Tfh Phenotype

AITLs and PTCLs with the Tfh phenotype are grouped together, because they share Tfh cells as their putative cell of origin and gene expression patterns [5]. For a PTCL to be designated as having the Tfh phenotype, its cells must express at least two antigens related to Tfh, which include PD1, BCL6, CD10, CXCL13, ICOS, and CXCR5. GEP studies have revealed that certain oncogenic pathways are enriched in AITL, including the NF-κB, IL6, and TGF-β signaling pathways [7,15]. Cytogenetic studies have demonstrated that AITL frequently also harbors trisomy 5, which is often accompanied by trisomy 21. Furthermore, small focal gene deletions can dysregulate the PI3 K-AKT-mTOR pathway [4,7].

Mutations of the TCR signaling pathway have been identified in AITLs and PTCLs with the Tfh phenotype. Fusions of ITK, including ITK-FER and ITK-SYK fusions, have been found in some of these tumors [10]. Costimulatory molecules of the TCR pathway are essential for an immune response, and mutations in CD28, PI3 K, PLCG1, CTNNB1, FYN, and GTF2 I have been identified. Two structural changes that result in CD28-ICOS and CD28-CTLA4 fusion proteins hyperactivate CD28 signaling and increase T-cell activation [16]. CD28 mutations in AITL are associated with a poor prognosis [17].

The activation of TCR signaling also activates GTPases, one of which is RHOA. The G17 V mutation in RHOA occurs in approximately 70% of AITLs exclusively within a background of TET2 mutations and independently of IDH2 mutations [7,18]. RHOA is part of the RHOA-VAV1 signaling pathway. When RHOA harbors the G17 V mutation, it recruits VAV1 and augments its adapter function, which accelerates TCR signaling. In preclinical models, the G17 V mutation has been shown to induce Tfh cell differentiation, upregulate ICOS expression, and increase PI3 K and MAPK signaling. An isolated VAV1 mutation and a VAV1-STAP2 translocation have also been identified in AITL [17,19].

AITLs and PTCLs with the Tfh phenotype have a high degree of epigenetic dysregulation, where the mutations TET2, IDH2, and DNMT3A have been detected in these subtypes. TET2—inactivating mutations occur in up to 85% of AITLs [20]. TET2 mutations cause DNA hypermethylation, affect other proteins such as histone deacetylase 1 (HDAC1) and HDAC2, and measurably decrease 5-hydroxymethylcytosine, as detected by immunohistochemistry. They are also correlated with the Tfh phenotype and poor clinical outcomes [7,20]. Mutations in DNMT3A, which is responsible for de novo DNA methylation, result in hypermethylation. The loss of both DNMT3A and TET2 drives the lymphoid transformation of Tfh cells in mouse models. Similar alterations of TET2 and DNMT3A have been found in AITL and myeloid neoplasm precursor cells, which suggests that these alterations are premalignant and, paired with additional defects, cause malignant transformations [10,21]. In addition, approximately 30% of AITLs bear IDH2 mutations at the R172 position, producing elevated levels of 2-hydroxyglutarate, which may act as an oncometabolite by inhibiting DNA hydroxylases and histone lysine-demethylases of the TET family [22]. IDH2 mutations are associated with AITL and are significantly related to Tfh signatures. TET2 and IDH R172 mutations often occur simultaneously in AITL, which suggests that they have a synergistic effect that promotes oncogenesis [23,24].

The tumor microenvironment may also play a significant role in T-cell lymphomagenesis. AITL frequently has B2M mutations, which protect tumor cells from cytotoxic T-cell related damage. As discussed above, the mutation and dysregulation of the costimulatory protein CD28 play a role in AITL oncogenesis. The upregulation of FLIP, an antiapoptotic regulatory protein, has also been documented in AITL. These mutations decrease the tumor cell immunogenicity and also provide survival and proliferation benefits [17,25,26]. AITLs and their related Tfh-like entities are known to overexpress cytokines that modify antitumor responses (e.g., interleukin (IL)-17, IL-10, IL-6, and IL-2) and angiogenesis (IL-8). Incompetent regulatory T cells (Tregs), which are functionally impaired and indirectly favor tumor growth, dampen the antitumor immune response in AITL [27]. The upregulation of PD-L1, a checkpoint protein, has also been identified in AITLs and other PTCLs. Intratumor endothelial cells, which produce excessive VEGF (which, in turn, promotes tumor cell growth), could explain the high vascular density in AITL [17,28].

Epstein–Barr virus (EBV)-infected B cells have been found in AITL; however, their role in its pathogenesis is not clear. Owing to the immunosuppressive environment in AITL, EBV-infected B cells might proliferate into diffuse large B-cell lymphomas. Interestingly however, EBV-negative diffuse large B-cell lymphomas have also been reported in patients with Tfh-cell lymphomas [10,29].

### 3.2. PTCL-NOS

GEP studies have subdivided PTCL-NOS into two main subgroups. One subgroup has upregulation of the transcription factor GATA3 and its downstream targets (e.g., CCR4, CXCR7, IK, and IL18 RA). This subgroup is associated with a poor prognosis, has a high proliferation signature, and lacks a strong tumor microenvironment signature. The GATA3 subgroup is associated with a greater genomic complexity, as evidenced by the presence of mutations of tumor suppressor genes in the PTEN-PI3K and CDKN2AB-TP3 pathways and augmented activity of MYC and STAT3. The other subgroup has a high expression of TBX21 and EOMES and their target genes (e.g., CXCR3, IL2RB, CCL3, and IFN-γ). The TBX21 subgroup is generally associated with more favorable outcomes and displays enrichment of the NF-κB pathway. A small subset of the TBX21 subgroup has a frequent CD8 expression and a cytotoxic signature; this subset is associated with poor clinical outcomes. GATA3 regulates the differentiation of T-helper type 2 cells, and TBX21 regulates the differentiation of T-helper type 1 cells and cytotoxic T cells. Furthermore, the two subtypes have significantly different copy number aberrations, indicating that they each have a different cell of origin [4,7,10,13].

One distinct molecular subtype of PTCL-NOS that lacks the Tfh phenotype primarily harbors TP53 and CDKN2A mutations. These alterations are inversely related to the Tfh phenotype in PTCL-NOS. Copy number profiling has identified extensive chromosomal abnormalities in this subtype, which has frequent alterations in genes related to immune surveillance. In addition, the subtype demonstrates frequent somatic mutations in transcription factors (IKZF2, ETV6, and PRDM1); transcriptional corepressors (IRF2BP2 and TBL1XR1); and RNA-binding proteins (YTHDF2 and DDX3X) [30].

Mutations and copy number variations in components of the TCR and NF-κB pathways, including in PLCG1, CARD11, CD28, VAV1, FYN, PTPRC, and TNFAIP3, have been detected in PTCL-NOS [10]. SYK, which is downstream of the TCR, is constitutively expressed in some cases of PTCL-NOS. The translocation t (5; 9) (q33; q22) results in ITK-SYK fusion, which results in SYK overexpression [31]. Constitutive NOTCH signaling, which could be attributed to genomic defects or a microenvironment-mediated mechanism, occurs in more than half of PTCL-NOS cases [32]. In mouse models, the tumorigenic activity of NOTCH1 involves the loss of TCF-1 and subsequent upregulation LEF-1 [33]. Interestingly, TCF-1 and LEF-1 losses have been detected primarily in PTCLs with the T-helper type 2 phenotype. Inactivating RHOA mutations have been noted in PTCL-NOS [17]. In addition, PTCL-NOS is known to carry VAV1 alterations in which VAV1 is translocated to form gene fusions, including VAV1-THAP4, VAV1-MYO1F, and VAV1-S100 A7, which lack the C-terminal SH3 domain and induce the constitutive action of VAV1 [34]. PTCL-NOS also has enriched PI3K pathway signatures. Frequent mutations of the tumor suppressor FAT1 have been identified in PTCL-NOS and are associated with poorer outcomes. Mutations in the tumor suppressor genes TP53, TP63, LATS1, STK3, and ATM have also been detected, albeit at lower frequencies [35].

Epigenetic alterations occur less frequently in PTCL-NOS than in AITL. PTCL-NOS is known to carry multiple mutations in epigenetic regulators, including MLL, MLL2, KDM6A, TET2, DNMT3A, SETD1B, SETD2, KMT2C, KMT2D, CREBBP, ARID1A, and EP300. Mutations in MLL, MLL2, and KDM6A are related to a poor prognosis [7,10,35].

The tumor microenvironment has an important role in the pathogenesis of PTCL-NOS. B2M mutations protect tumor cells against damage mediated by cytotoxic T cells [25]. In PTCL-NOS with TP53 or CDKN2A alterations, mutations that lead to immune evasion have been documented in HLA-A, HLA-B, CIITA, PD-L1, CD58, FAS, and PDCD1 [17,30]. The overexpression of the antiapoptotic protein FLIP has been documented in PTCL-NOS [26]. PTCL with CD47 upregulations, which inhibit antitumor macrophage activity, have also been reported [17,36]. Non-neoplastic FOXP3-positive suppressor Tregs, which sustain tumor cell survival, have been associated with the PTCL-NOS microenvironment [27]. Tumor-associated macrophages and relevant gene signatures have been recognized in PTCL-NOS and other subtypes. Generally, increased numbers of tumor-associated macrophages are correlated with a poor prognosis, indicating that these cells have a probable role in oncogenesis [37,38].

### 3.3. Anaplastic Large Cell Lymphoma

Anaplastic large cell lymphoma (ALCL) is characterized by large anaplastic cells strongly positive for CD30. CD30 belongs to the tumor necrosis factor receptor family and exerts diverse effects on cell growth and survival, chiefly through the activation of the NF-κB pathway. ALCLs are divided into two subgroups based on whether they express anaplastic lymphoma kinase (ALK). ALK-positive ALCL is defined by the formation of ALK fusion proteins; the NPM1-ALK fusion caused by translocation t(2; 5)(p23; q35) is characteristic of ALK-positive ALCL [39,40]. ALK-positive ALCL with TRAF1-ALK fusion displays a loss of TP53 and PRDM1, activation of NF-κB, and an aggressive clinical course [41]. These fusion proteins have constitutive tyrosine kinase activity and, consequently, activate downstream pathways.

STAT3 activity is enhanced due to ALK fusion proteins in ALK-positive ALCL and activating alterations in the STAT3 gene in ALK-negative ALCL [42]. Functional losses of phosphatases encoded by PTPN2 and PTPN6 have been detected in ALCL. Excessive STAT3 phosphorylation due to PTPN6 loss is related to shorter overall survival (OS) durations [10]. In ALK-negative cell lines, STAT3 and JAK1 are the most commonly mutated, both of which are associated with an adverse effect on survival [43]. Both ALK-negative ALCLs and ALK-positive ALCLs have a high expression of BATF3 and TMOD1 and low expression of TCR signaling-related genes, including CD3ε, ZAP70, LAT, and SLP76 [7,40]. In addition to the ALK status, NF-κB pathway signatures carry prognostic significance in ALCL [41].

Compared with ALK-positive ALCLs, ALK-negative ALCLs have more complex genomic alterations. The recurrent rearrangements of DUSP22 and TP63 have been documented in ALK-negative ALCLs. These translocations are mutually exclusive. DUSP22 translocation, which inhibits TCR signaling and promotes apoptosis, is associated with DNA hypomethylation, low PD-1 expression, high CD58 and HLA class II expression, and a very favorable prognosis. TP63 translocation is correlated with poor survival. The loss of PRDM1 and TP53, which are more frequent in ALK-negative ALCL than in ALK-positive ALCL, carries a poor prognosis [44,45]. The JAK-STAT3 pathway in ALK-negative ALCL is overactive owing to JAK1 and STAT3 mutations or fusion proteins of ROS1, TYK2, NFKB2, and NCOR2. GEP studies have demonstrated that ALK-negative ALCL has enriched IRF4 and MYC signatures [7,42]. Some ALK-negative ALCLs simultaneously express COL29 A1 and truncated forms of ERBB4 and, thus, might constitute a new subgroup [46]. In some ALCLs, FLIP overexpression reduces the immunogenicity, and suppressor Tregs, increased PD-L1 expression, and tumor-associated macrophage-related immunoregulation in the microenvironment play a role in tumor progression [17].

### 3.4. Adult T-Cell Leukemia/Lymphoma

Adult T-cell leukemia/lymphoma (ATLL) is associated with human T-cell lymphotropic virus type 1 (HTLV1) infection. HTLV1 codes for multiple proteins; among these, TAX1 contributes to viral transcription and the activation of signaling pathways, including the AP-1 and NF-κB pathways. The dysregulation of many tumor suppressor genes that promote genomic instability and cancer cell proliferation, including CDKN2A, CDKN2B, CDKN2C, RB, and TP53, has been documented in ATLL. ATLL does not express TAX1 in its later stages, but it persistently expresses the basic leucine zipper transcription factor HBZ [10,47]. HBZ expression stimulates cancer cell proliferation by inhibiting apoptosis and promoting the transcription of TERT and oncogenic miRNAs, including miR7 and miR21 [48,49]. It also inhibits CREBBP and KAT7, which are involved in p53 activation. HBZ also promotes tumor escape by upregulating TIGIT, a negative checkpoint inhibitor, and enhances the expression of IL-10 [10,50].

Of the NF-κB and TCR pathway mutations in ATLL, the most common are PLCG1, PRKCB, and CARD11 mutations. Hotspot mutations in FYN, VAV1, and IRF4 have also been reported [51]. CD28 signaling in ATLL is heightened by focal gains, hotspot mutations, and fusions of CD28 with CTLA4 or ICOS. Mutations in the receptors involved in T-cell trafficking, including CCR4 and CCR7, have also been found, as have mutations in the JAK-STAT pathway, including in STAT3, JAK1, and JAK3 [10,51]. Activating mutations in NOTCH1 occur in approximately 30% of ATLL cases, whereas the expression of FBXW7 mutations occur in 25% of cases [52]. TACC3, vital for microtubule growth during cell division, is an independent prognostic factor for poor OS [53]. The hepatocyte growth factor may activate the c-Met pathway in an autocrine manner and confer aggressiveness to lymphomatous ATLL [54].

Alterations of DNA methylation genes, such as TET2 and DNMT3A, occur far less frequently in ATLL than in AITL. CpG island hypermethylation in ATLL is often found, and extensive hypermethylation is associated with a poor OS. Polycomb-dependent repression is enriched by the trimethylation of histone 3 lysine 27 (H3K27me3) in ATLL. In association with H3K27me3, the expression of EZH2 and other PRC2 components is upregulated, and the expression of BIM, CDKN1A, CD7, and KDM6B is significantly downregulated in ATLL [7,55]. Mutations in ATLL that help tumor cells avoid immune surveillance include mutations in B2M, HLA-A, HLA-B, CD58, FAS, and PD-1, as well as those interfering with TRAIL-mediated apoptosis and IAP expression [17].

### 3.5. Extranodal NK/T-Cell Lymphoma

Extranodal NK/T-cell lymphoma (ENKTL) is frequently associated with EBV infections. However, the role of EBV in the origin and pathobiology of the disease is not clear. Some researchers have postulated that the neoplastic proliferation arises from the buildup of genomic defects in EBV-positive NK cells or T cells [56,57]. Recurrent nucleotide polymorphisms of the MHC class II gene loci on chromosome 6 are associated with ENKTL; of these polymorphisms, those located in the HLA-DPB1 region are the most strongly associated with the disease. Tumor cells express EBV proteins, including LMP1, LMP2, EBNA1, and EBERs. These proteins have a role in B-cell transformation, but their role in ENKTL has not been confirmed [10,58].

Immunohistochemistry revealed the overexpression of PDGFRα in ENKTL, indicating the activation of the PDGFR pathway. GEP detected the activation of the NOTCH pathway in the disease [59]. The TP53 pathway in ENKTL is likely to be dysregulated through multiple mechanisms. The miRNAs miR-BART8 and miR-BART20-5p decrease the TP53 expression by repressing the IFN-STAT1 pathway [60]. Upregulation of the mitosis regulator Aurora A kinase downregulates TP53 while increasing the BIRC5 expression. EBNA1 also decreases the TP53 expression. Repeated gains and losses of chromosomal copy numbers, including those encoding PRDM1, HACE1, ATG5, and AIM1, have been reported [7,61]. Mutations in the JAK-STAT pathway in ENKTL include mutations in JAK3, STAT3, and STAT5B. STAT5B mutations are relatively less common and are associated with a poor prognosis [17,62].

Mutations in the epigenetic modifiers MLL2, ARID1A, ASXL3, and EP300 have also been detected in ENKTL. Pro-apoptosis genes such as BIM, DAPK1, and DDX3X are downregulated. ASNS, which codes for asparaginase synthetase, is frequently methylated. FAS mutations and deletions have also been found in ENKTL. The upregulation of BCLXL and BCL2 via the activation of STAT5 or STAT3 may help ENKTL cells evade apoptosis. ENKTL cells have increased IL-10 and VEGF secretion, which encourages an immunosuppressive environment [7,8,10].

## 4. Role of CD30

CD30, a member of the tumor necrosis factor receptor family, is a transmembrane protein with a restricted expression on activated B and T cells in normal or inflamed tissue. Among the PTCL subtypes, CD30 is expressed strongly in ALCL but has variable expression across other subtypes, including PTCL-NOS, AITL, ATLL, and ENKTL [63]. CD30 stimulation leads to signal mediation via TRAF proteins to activate the NF-κB, MAPK, and ERK pathways, thereby exerting pleiotropic antiapoptotic and pro-survival effects. In addition, MAPK/ERK pathways activate the transcription factor JunB, which contributes to pro-survival effects and upregulates CD30 [64].

CD30-positive PTCLs may share phenotypic and molecular signatures. However, CD30-positive PTCLs and CD30-negative PTCLs have significant differences in their expression of proximal TCR signaling, T-cell activation, and differentiation molecules. The expression of the tyrosine kinases LCK, FYN, and ITK (as well as that of antigens CD52, CD69, and ICOS and the transcription factor NFAT) is largely conserved in CD30-negative cases but absent or prominently reduced in CD30-positive cases. Other transcription factors, including JunB and MUM1/IRF4, are significantly upregulated in CD30-positive cases as compared with CD30-negative cases [65].

The importance of CD30 expression on PTCL cells has not yet been completely defined. Indeed, after CD30 ligation, the NF-κB, MAPK, and ERK pathways may provide proliferation and survival benefits to tumor cells. Whether CD30 plays a role in malignant transformation, the creation of a tumor-sustaining microenvironment or a combination of effects is not clear. Nevertheless, the CD30 overexpression in certain subtypes warrants its investigation as a therapeutic target in PTCL.

## 5. Current Treatment Standards of Care

Most PTCL patients receive induction chemotherapy with CHOP (cyclophosphamide, doxorubicin, vincristine, and prednisone) or CHOP-like regimens [66]. Although chemotherapy with CHOP initially elicits a response in many patients, few patients achieve complete remissions, and many of these remissions are not durable [67]. Unfortunately, efforts to investigate the efficacy of novel agents in combination with CHOP have had limited success.

Etoposide is sometimes added to CHOP in patients who are older than 60 years and have normal lactate dehydrogenase levels. In one study, etoposide plus CHOP provided an event-free survival advantage over CHOP alone in these patients (75.4% vs. 51.0%) [68]. However, another study with a large sample population (1933 patients) concluded that the addition of etoposide to CHOP-like regimens for frontline therapy does not improve survival in PTCL patients but, rather, leads to prolonged cytopenia and hospitalization [69]. Regardless of the type of frontline treatment they receive, most patients have disease relapses. Patients whose disease progresses or relapses after first-line therapy have very poor survival outcomes [66].

The randomized phase III ECHELON-2 trial, which enrolled 452 untreated patients with CD30-positive PTCL, compared the efficacy of CHOP to that of brentuximab vedotin (BV) plus CHP (cyclophosphamide, doxorubicin, and prednisone). Compared with the CHOP group, the BV-plus-CHP group had a significantly higher overall response rate (ORR; 83% vs. 72%), complete remission rate (CRR; 68% vs. 56%), and 3-year progression-free survival (PFS) rate (57.1% vs. 44.4%). Based on these results, the U.S. Food and Drug Administration (FDA) approved BV plus CHP for use as a first-line therapy for PTCL [70].

An open-label prospective trial conducted in China compared CHOP and GDPT (gemcitabine, cisplatin, prednisone, and thalidomide) as treatments for 153 patients with newly diagnosed PTCL. Compared with the CHOP group, the GDPT group had a significantly higher ORR (66.3% vs. 50%; *p* = 0.042) and CRR (42.9% vs. 27.6%; *p* = 0.049). At a median follow-up of 2 years, the GDPT group also had a significantly higher PFS rate (57% vs. 35%; *p* = 0.0035) and OS rate (71% vs. 50%; *p* = 0.0001) [71].

Conventional CHOP-like therapy has not elicited the desired responses for patients with ENKTL, but asparaginase-based combinations may prove to be effective. Although the standard frontline therapy has not been established, the efficacies of regimens such as SMILE (dexamethasone, methotrexate, ifosfamide, L-asparaginase, and etoposide) and DDGP (cisplatin, dexamethasone, gemcitabine, and pegaspargase) are being evaluated in clinical trials. A randomized multicenter study compared the efficacy and survival outcomes of 80 patients with newly diagnosed stage III/IV ENKTL who were treated with DDGP or SMILE. The ORR of the DDGP group was significantly higher than that of the SMILE group (90% vs. 60%; *p* = 0.002). In addition, compared with SMILE patients, DDGP patients had a significantly higher 3-year PFS rate (56.6% vs. 41.8%; *p* = 0.004) and 5-year OS rate (74.3% vs. 51.7%; *p* = 0.02) [72].

Autologous stem cell transplantation (ASCT) as consolidation following induction therapy for PTCL is currently debatable due to a paucity of randomized trial data. However, nonrandomized studies have suggested that ASCT following the response to initial therapy improves its survival. One large study by the Nordic Lymphoma Group, which enrolled 160 patients with untreated PTCL (all subtypes except ALK-positive ALCL), evaluated the role of high-dose chemotherapy (HDT) and ASCT as consolidation therapy in patients who responded to CHOP or CHOEP (cyclophosphamide, hydroxydaunorubicin, vincristine, etoposide, and prednisone). A total of 115 patients received high-dose chemotherapy and ASCT; the 5-year PFS and OS rates were 44% and 51%, respectively. Two transplant-related deaths occurred, and 28 patients experienced progression or relapse within 2 years of transplantation [73]. In a retrospective analysis of 58 PTCL patients, 40 of whom received upfront ASCT, the estimated 5-year PFS and OS rates were 35% and 41%, respectively. ASCT was well-tolerated, and no transplant-related deaths occurred; however, 48% of the patients receiving ASCT had a disease relapse after the transplant [74]. Findings from a multivariate analysis in another prospective study that compared the survival of PTCL patients in the first complete remission with or without ASCT suggested that ASCT is independently associated with improved survival. Of 119 patients with PTCL (ALK-negative ALCL, AITL, or PTCL-NOS) who were in the first complete remission, 36 underwent ASCT. At a median follow-up of 2.8 years, the median OS duration of the non-ASCT group was 57.6 months, whereas that of the ASCT group had not been reached [75].

The FDA has approved only four drugs for the treatment of patients with relapsed or refractory (R/R) PTCL: romidepsin, belinostat, pralatrexate, and BV [76,77,78]. Belinostat and romidepsin are HDAC inhibitors, pralatrexate is an antifolate agent, and BV is an anti-CD30 monoclonal antibody conjugated to a microtubule inhibitor. Compared with ALCL patients treated with BV, PTCL patients have had low response rates with these agents, with short PFS durations and no significant change in the OS durations. Furthermore, apart from CD30, which serves as a biomarker of the BV response, no other predictive biomarkers of response have been identified [66].

## 6. Novel Targeted Therapies

This section highlights the novel targeted agents being studied alone or in combination with other drugs for the treatment of PTCL. Some of these drugs, along with their molecular targets, are included in Figure 1. The results from the clinical trials evaluating these agents are summarized in Table 2.

### 6.1. T-Cell and Costimulatory Receptor Signaling Inhibitors

The TCR and costimulatory receptor pathways, which are often altered in PTCL, are potential therapeutic targets. In one study, eight out of 12 AITL patients had a response to cyclosporine A, a calcineurin inhibitor that blocks TCR signaling, and three of these patients had a complete remission [93]. VAV1 activation accelerates TCR signaling, and dasatinib, a multikinase inhibitor, has been shown to block VAV1 activation in vitro and improves OS in mouse models. Dasatinib targets the Src kinase and, thus, could be therapeutically targeted in PTCLs bearing FYN mutations. Mutations in CD28, a costimulatory receptor, are correlated with a poor prognosis, and CD28-CTLA4 fusion has been identified in PTCL. Thus, ipilimumab, an anti-CTLA4 monoclonal antibody, could be studied as targeted therapy [16].

### 6.2. PI3K Inhibitors

The PI3Kγ and PI3Kδ isoforms are often required for the TCR-mediated activation of T cells. Duvelisib, an inhibitor of both PI3K isoforms, elicited an ORR of 50% and CRR of 18.8% in a phase I trial enrolling 16 patients with R/R PTCL. The most common grade 3 or 4 adverse events (AEs) were transaminase increases, neutropenia, and maculopapular rash [94]. Duvelisib, in combination with either romidepsin or bortezomib for the treatment of R/R PTCL, is currently being studied in a phase Ib/II trial. A preliminary analysis revealed that, in 27 patients, duvelisib plus romidepsin elicited an ORR of 59% and CRR of 33%, with tolerable side effects. Interestingly, a subset of eight AITL patients had an ORR of 75% and CRR of 63%. Frequent grade 3 or 4 AEs were neutropenia, elevated transaminase levels, and hyponatremia [79]. Another dual PI3Kγ/δ inhibitor, tenalisib, was found to be tolerable and had encouraging efficacy in a phase I/Ib study enrolling patients with R/R PTCL or CTCL. For 35 evaluable patients, the ORR was 45.7%. Among the PTCL patients, the ORR was 46.7%, and the CRR was 20%. Common AEs were fatigue, elevated transaminase levels, and diarrhea [80].

### 6.3. Proteasome Inhibitors

The proteasome inhibitor bortezomib, which inhibits NF-κB, has shown promise in the treatment of ATLL. In a phase II trial of bortezomib in 15 patients with R/R ATLL, one patient had a partial remission, and five patients had a stable disease [81]. All patients experienced one or more AEs, the most common of which were fever and thrombocytopenia. Grade 3 or higher AEs included thrombocytopenia, lymphopenia, leukopenia, and peripheral neuropathy [81]. Other trials are investigating the use of bortezomib combined with CHEP (cyclophosphamide, doxorubicin, etoposide, and prednisone) for untreated PTCL and the use of a similar agent, ixazomib, combined with romidepsin for R/R PTCL [16].

### 6.4. JAK-STAT Pathway and SYK Inhibitors

The JAK-STAT pathway is a potential therapeutic target in PTCL. The JAK inhibitor ruxolitinib in the treatment of PTCL and CTCL is under investigation. In one study that enrolled 33 patients with PTCL or CTCL, of whom 27 were evaluable, ruxolitinib elicited a 30% ORR and 4% CRR. Grade 3 or more drug-related AEs included neutropenia, lymphopenia, anemia, and thrombocytopenia [82]. SYK, a receptor tyrosine kinase, is another promising target, as it is expressed in 94% of PTCLs [83]. In preclinical studies, the SYK inhibitor R406 efficiently initiated apoptosis and repressed proliferation in T-cell lymphomas [95]. Cerdulatinib, a dual SYK/JAK inhibitor, has also exhibited substantial efficacy and tolerability in a phase IIa clinical trial in patients with R/R PTCL, eliciting an ORR of 35% and CRR of 31%. Frequently observed AEs were diarrhea, nausea, neutropenia, and elevated lipase and amylase levels [90].

### 6.5. mTOR Inhibitor

An inhibitor of the mTOR pathway, everolimus, has significant activity against the proliferation of malignant T cells in vitro. In a phase II study of everolimus in 16 patients with R/R PTCL or CTCL of different subtypes, the ORR was 44%, and the median PFS and OS durations were 8.5 months and 10.2 months, respectively. Among the patients whose disease responded to everolimus, the median response duration was 8.5 months. Grade 3 or 4 hematologic toxicities that were possibly related to everolimus included anemia, leukopenia, neutropenia, and thrombocytopenia. Nonhematologic grade 3 or more toxicities reported with this agent included hyperglycemia, hypertriglyceridemia, and hypercholesterolemia [84].

### 6.6. Epigenetic Modulators

DNA methylation genes are mutated in multiple PTCL subtypes and are frequently mutated in AITL. Therefore, hypomethylating drugs and HDAC inhibitors could prove to be effective against these diseases. In one study, a group of patients with R/R AITL had sustained responses to the hypomethylating agent azacytidine, with an ORR of 75% and CRR of 50%. The drug was well-tolerated, and only one patient developed grade 3 diarrhea [85]. Moreover, investigators have reported that combinations of hypomethylating drugs and HDAC inhibitors have a marked activity against PTCL. In a phase I multicenter study in PTCL patients, the combination of azacytidine and romidepsin was well-tolerated, eliciting an ORR of 73% and CRR of 55%. The dose-limiting toxicities seen were grade 3 or more thrombocytopenia and neutropenia [86]. Azacytidine is currently being investigated in combination with CHOP, pralatrexate, durvalumab, and bortezomib in clinical trials [10].

Chidamide, another HDAC inhibitor, for the treatment of R/R PTCL was investigated in a phase II study in China. The study enrolled 83 patients with AITL, ALCL, PTCL-NOS, or NKTCL. The ORR was 28%, and the CRR was 14%. Although the median PFS duration was only 2.1 months, the median OS duration was 21.4 months. AITL patients had the highest ORR (50%) and CRR (40%). Most AEs were grade 1 or 2 events. Grade 3 or more AEs were thrombocytopenia, leucopenia, and neutropenia [87].

### 6.7. Aurora A Kinase Inhibitor

Aurora A kinase, which is overexpressed in aggressive lymphomas, plays a vital role in mitosis during cell cycle progression. In the phase III LUMIERE trial, the Aurora A kinase inhibitor alisertib was compared with a choice of romidepsin, pralatrexate, or gemcitabine. Patients receiving alisertib had an ORR of 33%, a CRR of 18%, and median PFS and OS durations of 3.8 months and 13.7 months, respectively. An improved response with alisertib was, however, not demonstrated. Frequent grade 3 or more AEs related to alisertib included neutropenia, anemia, thrombocytopenia, stomatitis, and diarrhea [88]. However, the combination of romidepsin and alisertib was investigated in a phase I study that included four patients with R/R PTCL but elicited no response [89].

### 6.8. ALK Inhibitor

The ALK1 inhibitor crizotinib was investigated in 26 pediatric patients with R/R ALK-positive ALCL. In the two dosage groups evaluated, the ORRs were 83% and 90%, respectively, and the CRRs were 80% and 83%, respectively. Neutropenia was the most frequently reported grade 3 or more AE [96]. ALK-positive ALCL, which is resistant to crizotinib, could be treated with a blockade of platelet-derived growth factor receptor-β (PDGFRB), which has been shown to encourage lymphoma development and tumor propagation in mouse models of ALCL with NPM-ALK fusions. Imatinib, which inhibits both PDGFRB and PDGFRA, was reported to induce a complete remission in a patient with refractory ALK-positive ALCL [97]. Recently, the FDA approved crizotinib in children and young adults with relapsed/refractory ALCL after the results from a trial with 26 patients [98].

### 6.9. Anti-CCR4 Monoclonal Antibody

An anti-CCR4 monoclonal antibody, mogamulizumab, has elicited effective antitumor responses in PTCL cell lines and mouse models of ATLL [99]. In Japan, the agent is approved for the treatment of ATLL based on the results of a multicenter phase II study, which revealed the agent’s encouraging efficacy and tolerable side effects in patients with relapsed aggressive CCR4-positive ATLL. For the 28 patients enrolled in the study, the ORR was 50%, the CRR was 31%, and the PFS and OS durations were 5.2 months and 13.7 months, respectively. The most frequent AEs were skin rashes and infusion reactions, which were manageable. Most grade 3 or 4 AEs were hematologic and included lymphopenia, leukopenia, neutropenia, and thrombocytopenia [100]. The FDA has approved mogamulizumab for the treatment of R/R mycosis fungoides and Sezary syndrome but not for the treatment of PTCL-NOS [10].

### 6.10. Anti-CD25 Antibody

Camidanlumab tesirine (ADCT-301) is a conjugate of an anti-CD25 (also called IL-2 receptor subunit α) antibody and a pyrrolobenzodiazepine dimer toxin (tesirine) whose internalization through the IL-2 receptor results in cell death [101]. A multicenter phase I trial of camidanlumab tesirine enrolled 44 patients with R/R NHL, including 22 with T-cell lymphoma. For the 19 evaluable T-cell lymphoma patients, the ORR was 42.1%, and the CRR was 5.3% [102].

### 6.11. Anti-CD52 Antibody

Multiple PTCL subtypes express CD52, a glycoprotein present on the surfaces of T- and B-lymphocytes and NK cells. Although such expression warrants the investigation of alemtuzumab, an anti-CD52 antibody, for the treatment of PTCL, the expression of CD52 on normal T cells and B cells has limited the potential therapeutic use of alemtuzumab and raises concerns of a possible immunosuppression. Nevertheless, researchers conducted a pooled analysis of two phase III trials of CHOP plus alemtuzumab versus CHOP alone in 252 untreated PTCL patients. The CRR of the CHOP-plus alemtuzumab group (56%) was higher than that of the CHOP-alone group (43%), but the combination was not found to confer a benefit in the PFS and OS. Grade 3 or 4 hematologic toxicities were more common in the alemtuzumab group and included leukopenia, anemia, and thrombocytopenia [89,103].

### 6.12. Immunomodulator

An immunomodulatory drug, lenalidomide, has activity in the therapy of B-cell NHL. In the phase I/II EXPECT trial of lenalidomide of 54 patients with R/R PTCL, the. patients were treated with a median of three prior therapies. For all patients, the ORR was 22%, and the CRR was 11%. AITL patients had a higher ORR (31%) and CRR (15%). The PFS duration was 2.5 months overall but 4.6 months specifically among AITL patients. The common grade 3 or 4 hematologic toxicities were neutropenia and thrombocytopenia. Thirty-five percent of the patients experienced at least one AE that led to a dose reduction or interruption. Life-threatening AEs were reported in 54% of the patients; 12 patients died during the study [104].

### 6.13. Bispecific Antibodies

Molecularly engineered antibodies fall into two broad categories: immunotoxins and bispecific antibodies. The former has two arms; one that recognizes a tumor cell marker, and the other is bound to a toxin or drug. The latter are designed to recognize two molecular targets [105]. An anti-CD30/CD16 bispecific antibody, AFM13, directs NK cells against CD30-positive tumors. However, its efficacy against T-cell lymphomas has not been established [10]. Bispecific antibodies have also been designed to link a component of the TCR complex to a tumor surface marker, thereby directing T-cell cytotoxic activity towards the tumor [106]. A CD19/CD3 bispecific antibody, blinatumomab, yielded durable responses in patients with B-cell NHL, spurring interest in this emerging modality [107].

### 6.14. Checkpoint Inhibitors

PD-L1 and PD-1 are constitutively expressed in multiple PTCL subtypes and on host cells in the tumor microenvironment. The inhibition of these immune checkpoint molecules has shown a therapeutic efficacy against NHL [108]. In one retrospective study of seven patients with R/R NKTCL, pembrolizumab, an anti-PD-1 antibody, displayed high efficacy against the disease, with an ORR of 100% (7/7) and CRR of 71% (5/7). After a median of seven therapy cycles and a median follow-up of 6 months, all five patients who had complete remission still had no evidence of the disease [109]. In a phase II study in patients with R/R PTCL, pembrolizumab elicited modest activity, with an ORR of 33%, a CRR of 27%, and PFS and OS durations of 3.2 months and 10.6 months, respectively. The most common AE was a rash, and the most common grade 3 or higher AEs were a rash and pneumonitis [91]. A phase I/II study of a combination of pembrolizumab and romidepsin in 15 patients with R/R PTCL yielded early encouraging results, including an ORR of 44% and complete remission in three patients. Nausea, vomiting, and fatigue were the common grade 3 or more treatment-related AEs. Of note, two patients experienced hyper-progression of their disease within 10 days of treatment [92]. However, because PD-1 inhibits TCR signaling, there is some concern that its use in PTCL patients could result in lymphoma progression [66]. A report of rapid disease progression in ATLL patients receiving nivolumab supports this idea [110]. In one study, a patient developed PTCL secondary to treatment with a checkpoint inhibitor, possibly because PD-1 could also act as a tumor suppressor in T-cell lymphomas [111].

### 6.15. Inhibitors of Apoptosis

The inhibitor of apoptosis proteins (IAPs), which include cellular IAP (cIAP) and X-linked IAP (xIAP), have antiapoptotic functions and regulate cellular survival signaling pathways. XIAP inhibits caspases, which are involved in programmed cell death, and cIAP prevents the formation of proapoptotic signaling complexes [112]. The deregulation of IAP expression due to genetic alterations is seen in solid tumors and hematologic malignancies [113,114]. ASTX660, a novel non-peptidomimetic dual antagonist of cIAP1/2 and xIAP, has demonstrated clinical activity in a phase 1 clinical trial [115]. In a phase 2 trial, 16 peripheral T-cell lymphomas (PTCL) and 13 CTCL patients received ASTX at 180 mg/day on days 1–7 and 15–22 in a 28-day cycle. An ORR of 28% (by Lugano criteria) was seen in the 14 evaluable patients in the PTCL cohort, and three responding patients remained in the study. The most common AEs of any grade were subclinical lipase and amylase elevation. These results demonstrate the clinical activity of ASTX660 in T-cell lymphomas, and expansion of the phase 2 cohort is ongoing [116].

### 6.16. Chimeric Antigen Receptor T-Cell Therapy

Some researchers have encouraged the use of chimeric antigen receptor-T (CAR-T) cells as therapy for PTCL patients. Potential targets of such therapy include CD30, CD7, CD5, and CD4 [10]. Given the BV successful targeting of CD30, CD30 CAR-T cells seem particularly promising, and their effectiveness has been demonstrated in mouse models [117]. In a phase I trial in 18 patients with R/R Hodgkin’s lymphoma, CD30 CAR-T cells were shown to be safe, eliciting partial remission in seven patients and a stable disease in six patients [118]. In another phase I study of CD30 CAR-T cells in patients with CD30-positive disease, one of two patients with ALCL had a complete remission that lasted 9 months [119]. CAR-T cells targeting the TCR also seem promising. The effectiveness of CAR-T cells targeting the TCRβ chain has been documented [120]. CAR-NK cells have been investigated in preclinical studies, but clinical trials are required to validate their efficacy [121]. Developing CAR-T cells to target T-cell antigens seems promising; however, fratricide within a CAR-T-cell product might impede such a development. This issue might be addressed by using CRISPR technology to eliminate CAR-T cell targets in effector cells themselves [66].

## 7. Conclusions

Novel technologies such as genome sequencing and GEP have improved our understanding of the pathobiology of PTCL. Although the standard treatment of PTCL is still limited to only a handful of agents that have not yielded the desired clinical outcomes, genomic analyses have uncovered new therapeutic targets whose inhibitions might have efficacy against PTCL. However, aside from the ECHELON-2 trial, clinical trials of multiple agents given alone or in combination have not yielded significant breakthroughs in the treatment of newly diagnosed or R/R PTCL. Nevertheless, multiple combinations are being tested in the current clinical trials, and emerging research to further elucidate the initiation and progression of PTCL will continue to strengthen the efforts to identify new therapeutic options. Furthermore, the identification of prognostic and predictive markers would enable us to personalize treatment options. Although their use outside clinical trials is currently limited, we believe the future of PTCL treatment will center on targeted therapies that are supported by GEP.

## 8. Future Directions

Since studies have demonstrated that available single agents only have modest activity against PTCL, efforts are now being directed towards the investigation of combination regimens. In the future, GEP should be integrated into the diagnosis and management of PTCL to detect mutations and guide targeted therapy. Biomarkers that predict the response to targeted therapies will help us to choose effective therapies for individual patients. New data will help us choose treatment options with improved efficacy and survival benefits for PTCL patients, and patients should be encouraged to enroll in clinical trials. International efforts are needed to create large research networks and design multicenter randomized trials, which will be instrumental in conceiving rational treatment strategies and testing novel agents.

Practice points:An improved understanding of PTCL biology by GEP and genome sequencing has provided prognostic information and potential therapeutic targets.CHOP-like regimens remain the mainstay of frontline treatment, but the addition of newer agents to these regimens could improve the treatment efficacy and patient survival.The ECHELON-2 trial investigating the use of BV plus CHP for the frontline treatment of CD30-positive PTCL is the only trial to yield a significant breakthrough in PTCL treatment.Multiple novel targeted agents, alone or in combination, have shown promising efficacy and safety in PTCL patients.

Research agenda:Improve our understanding of PTCL biology and identify the prognostic subgroups within the PTCL subtypes.Undertake large multicenter trials of novel agents alone or in combination with other therapies to understand their efficacy and safety.Identify and validate the biomarkers that predict the PTCL response to novel targeted agents.

## Figures and Tables

**Figure 1 cancers-13-05627-f001:**
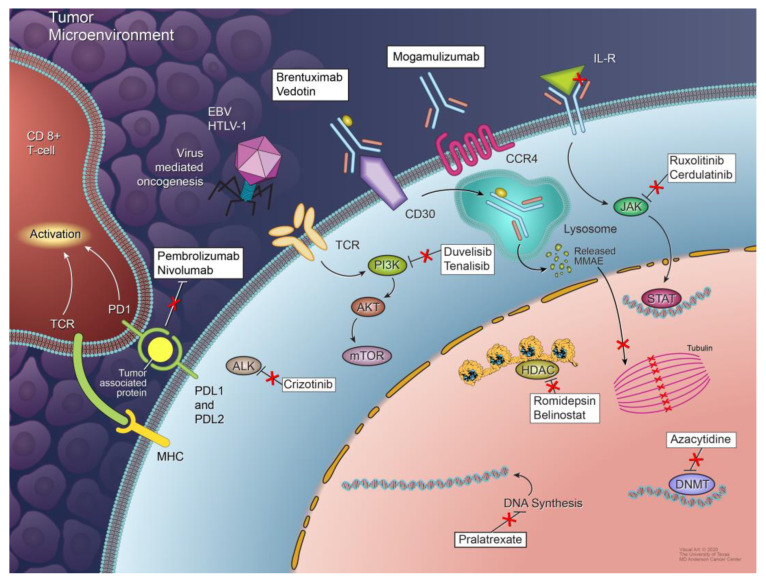
Mechanisms of peripheral T-cell lymphoma development and agents used for targeted therapies. Multiple mechanisms are responsible for the development and proliferation of different subtypes of PTCL, including signaling pathway deregulation, epigenetic dysregulation, tumor microenvironment signals, and virus-mediated oncogenesis. Not all of these mechanisms are involved in the pathogenesis of each PTCL subtype. Some potential therapeutic targets and agents targeting them are also shown.

**Table 1 cancers-13-05627-t001:** Common pathogenic mechanisms associated with peripheral T-cell lymphoma (PTCL) subtypes.

PTCL Subtype	Pathway	Molecular Event
AITL	TCR	Mutations: PI3 K pathway genes, *CD28*, *PLCG1*, *CTNNB1*, *FYN*, *GTF21*, *VAV1* Rearrangements: *CD28*
	RHOA	Mutations: *RHOA* Rearrangements: *VAV1*
	Epigenetic pathways	Mutations: *TET2*, *DNMT3 A*, *IDH2*
	Tumor microenvironment	Mutations: *B2 M*, *CD28* Upregulation: *FLIP*, *PD-L1* Cytokine- and Treg-mediated modulation
	Viral mechanisms	EBV-positive B cells (unclear role)
PTCL-NOS	TCR and NF-κB	Mutations: *PLCG1*, *CARD11*, *CD28*, *VAV1*, *FYN*, *PTPRC*, *TNFAIP3* Rearrangements: *SYK*
	NOTCH	Constitutive activation Loss: *TCF1*, *LEF1*
	PI3 K	Hyperactivation
	RHOA	Mutations: *RHOA* Rearrangements: *VAV1*
	Tumor suppressor genes	Mutations: *TP53*, *TP63*, *CDKN2 A*, *FAT1*, *ATM*
	Transcription regulation	Mutations: *IKZF2*, *ETV6*, *PRDM1*, *YTHDF2*, *DDX3 X*, *IRF2 BP2*, *TBL1 XR1*
	Epigenetic pathways	Mutations and/or CNAs: *MLL*, *KDM6 A*, *SETD1 B*, *SETD2*, *KMT2 C*, *KMT2 D*, *ARID1 A*, *EP300*
	Immune surveillance	Mutations: *HLA-A*, *HLA-B*, *CD58*, *FAS*, *CIITA*, *B2 M*, *PD-L1*
	Tumor microenvironment	Upregulation: FLIP, CD47 Cytokine-, Treg-, and TAM-mediated modulation
ALK-positive ALCL	TCR and CD30	NF-κB hyperactivation Rearrangements: *ALK* Low expression of TCR signaling—related genes, including CD3, ZAP70
	JAK-STAT	STAT3 activation Loss: *PTPN2*, *PTPN6*
	Tumor suppressors	Mutations or loss: *TP53*, *PRDM1* (with TRAF1-ALK fusion)
	Tumor microenvironment	Upregulation: PD-L1 Treg- and TAM-mediated modulation
ALK-negative ALCL	TCR and CD30	NF-κB hyperactivation Rearrangements: *DUSP22*, *TP63* Low expression of TCR signaling—related genes, including CD3, ZAP70
	JAK-STAT	Mutations: *JAK1*, *STAT3* Rearrangements: *ROS1*, *TYK2* Loss: *PTPN2*, *PTPN6*
	Tumor suppressors	Mutations or loss: *TP53*, *PRDM1*
	Tumor microenvironment	Upregulation: PD-L1 Treg- and TAM-mediated modulation
ATLL	Viral mechanisms	HTLV-1 has a significant role in pathogenesis
	TCR	Mutations: *PLCG1*, *CARD11*, *PRKCB* Rearrangements: *CD28*, *CTLA4*
	T-cell trafficking	Mutations: *CCR4*, *CCR7*
	JAK-STAT	Mutations: *JAK1*, *JAK3*, *STAT3*
	NOTCH	Mutations: *NOTCH1*, *FBXW7*
	Epigenetic pathways	Mutations: *TET2*, *DNMT3 A*, *EZH2*, *PRC2* CpG island hypermethylation
	Immune surveillance	Mutations: *HLA-A*, *HLA-B*, *CD58*, *FAS*, *B2 M*, *PD-L1*
ENKTL	Viral mechanisms	EBV-mediated pathogenesis
	JAK-STAT	Mutations: *JAK3*, *STAT3*, *STAT5 B*
	Tumor suppressors	Mutations: *TP53*, *PRDM1*
	Epigenetic pathways	Mutations: *MLL2*, *ARID1 A*, *ASXL3*, *EP300*

Abbreviations: AITL, angioimmunoblastic T-cell lymphoma; TCR, T-cell receptor; Treg, regulatory T-cell; EBV, Epstein-Barr virus; PTCL-NOS, peripheral T-cell lymphoma not otherwise specified; CNAs, copy number alterations; TAM, tumor-associated macrophages; ALK, anaplastic lymphoma kinase; ALCL, anaplastic large cell lymphoma; ATLL, adult T-cell leukemia/lymphoma; HTLV-1, human T-lymphotropic virus 1; ENKTL, extranodal natural killer/T-cell lymphoma.

**Table 2 cancers-13-05627-t002:** Results of clinical trials of targeted agents for the treatment of peripheral T-cell lymphoma.

Target	Therapy	Trial Phase	N	ORR	CRR	Survival	Grade ≥ 3 AEs	Ref
CD30	BV + CHP	III	226	83%	68%	3-year PFS: 57.1%	Neutropenia, anemia, diarrhea, peripheral neuropathy	[71]
PI3Kγ/δ	Duvelisib	I	16	50%	18.8%	mPFS: 8.3 months mOS: 8.4 months	Transaminase increases, neutropenia, maculopapular rash	[79]
PI3Kγ/δ and HDAC	Duvelisib + Romidepsin	I	27	59%	33%	NR	Neutropenia, elevated transaminases, hyponatremia	[80]
PI3Kγ/δ and Proteasome	Duvelisib + Bortezomib	I	16	44%	25%	NR	Neutropenia, elevated transaminases, hyponatremia	[80]
PI3Kγ/δ	Tenalisib	I/Ib	35 (PTCL and CTCL)	45.7%	8.6%	NR	Fatigue, elevated transaminases, diarrhea	[81]
Proteasome	Bortezomib	II	15 (ATLL)	6.7%	0%	PFS: 38 days	Thrombocytopenia, lymphopenia, leukopenia, peripheral neuropathy	[82]
JAK1 and JAK2	Ruxolitinib	II	27 (PTCL and CTCL)	30%	4%	NR	Neutropenia, lymphopenia, anemia, thrombocytopenia	[83]
SYK, JAK1, and JAK3	Cerdulatinib	IIa	26	35%	31%	NR	Lipase increases, amylase increases, neutropenia	[84]
mTOR	Everolimus	II	16 (PTCL and CTCL)	44%	NR	mPFS: 8.5 months mOS: 10.2 months	Anemia, leukopenia, neutropenia, thrombocytopenia, hyperglycemia, hypertriglyceridemia	[85]
DNA methylation	Azacytidine	Retrospective	12	75%	50%	mPFS: 15 months mOS: 21 months	Diarrhea	[86]
DNA methylation and HDAC	Azacytidine + Romidepsin	I	NR	73%	55%	NR	Thrombocytopenia, neutropenia	[87]
HDAC	Chidamide	II	83	28%	14%	mPFS: 2.1 months mOS: 21.4 months	Thrombocytopenia, leukopenia, neutropenia	[88]
AAK	Alisertib	III	138	33%	18%	mPFS: 3.8 months mOS: 13.7 months	Neutropenia, anemia, thrombocytopenia, stomatitis, diarrhea	[89]
ALK	Crizotinib	Retrospective	26 (ALK-positive ALCL)	83–90%	80–83%	NR	Neutropenia	[80]
CCR4	Mogamulizumab	II	26 (ATLL)	50%	31%	mPFS: 5.2 months mOS: 13.7 months	Lymphopenia, leukopenia, neutropenia, thrombocytopenia	[83]
CD25	Camidanlumab tesirine	I	19 (T-cell lymphoma)	42.1%	5.3%	NR	Exfoliative dermatitis, neuropathy, thyroiditis	[90]
CD52	Alemtuzumab + CHOP	III	123	NR	56%	3-year PFS: 33% 3-year OS: 46%	Leukopenia, anemia, thrombocytopenia	[84]
Immune modulation	Lenalidomide	I/II	54	22%	11%	mPFS: 2.5 months	Neutropenia, thrombocytopenia	[85]
PD-1	Pembrolizumab	Retrospective	7 (NKTCL)	100%	71%	NR	-	[91]
PD-1	Pembrolizumab	II	13	33%	27%	mPFS: 3.2 months mOS: 10.6 months	Rash, pneumonitis	[92]
Immune modulation	Etoposide + CHOP	Retrospective	609	-	NA	mPFS: 4.9 months 5-year survival: 17.9%	Anemia, thrombocytopenia	[69]
Immune modulation	Etoposide + CHOP	Retrospective	20	-	50%	mPFS: 9 months 5-year survival: 5%	Anemia, thrombocytopenia, neutropenia	[69]

Abbreviations: ORR, overall response rate; CRR, complete remission rate; AEs, adverse events; mPFS, median progression-free survival; mOS, median overall survival; NR, not reported, Brentuximab Vedotin; BV, (cyclophosphamide, doxorubicin, vincristine, and prednisone); CHOP.

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
