# Peer review of "Emerging Therapeutic Landscape of Peripheral T-Cell Lymphomas Based on Advances in Biology: Current Status and Future Directions"

_cancers, 2021, doi:10.3390/cancers13225627_

Round 1
Reviewer 1 Report
In review “Emerging Therapeutic Landscape of Peripheral T-Cell Lymphomas Based on Advances in Biology: Current Status and Future Directions” Melina khan and co-authors summarize the current therapeutic options for t-NHLs and provide a good and comprehensive overview of current treatment strategies.
Although the manuscript provides a in-depth review of the treatment option, substantial work should be done on the graphical representation and presentation of the data to provide a better overview of the complex PTCL therapeutic landscape. This includes both the graphical revision and different presentation of Table 1 (which is difficult to read/understand in its current form) and the revision/creation of a figure 1 which combines the pathogenesis of PTCLs (with deregulated pathways and mutation) with their respective drug(s).
Furthermore, following points need to be addressed:
- line11: “paper” has to be corrected to “review”
- table 1: Table 1 is – as mentioned above- very confusing. It needs to be graphically revised and also explained with a meaningful legend. Moreover, references need to be added.
- Figure 1: although some of the described drugs with their therapeutic target are displayed, the figure does not really add any insight how they mechanistically interact with the pathogenesis of the PTCL and should therefore be revised.
- paragraph 3.3: add "Lobello, C., Tichy, B., Bystry, V. et al. STAT3 and TP53 mutations associate with poor prognosis in anaplastic large cell lymphoma. Leukemia 35, 1500–1505 (2021). https://doi.org/10.1038/s41375-020-01093-1" reference
- line 280: sometimes PD-1 is reported as protein sometime as gene pcdc1, it has to be consistent
- abbreviation in the entire manuscript need to be controlled
- line 349: ref needs to be added as well as ref. to table2
- paragraph 6.8: FDA crizotinib approval for ALCL have to be mentioned here
Author Response
Response to Reviewer 1 Comments
Point 1: Although the manuscript provides a in-depth review of the treatment option, substantial work should be done on the graphical representation and presentation of the data to provide a better overview of the complex PTCL therapeutic landscape. This includes both the graphical revision and different presentation of Table 1 (which is difficult to read/understand in its current form) and the revision/creation of a figure 1 which combines the pathogenesis of PTCLs (with deregulated pathways and mutation) with their respective drug(s).
Response 1: The figure has been revised, and Table 1 has been restructured so that it is easily readable.
Point 2:
- line11: “paper” has to be corrected to “review”
Response 2: Change has been made.
Point 3:
- table 1: Table 1 is – as mentioned above- very confusing. It needs to be graphically revised and also explained with a meaningful legend. Moreover, references need to be added.
Response 3: Table 1 has been graphically revised.
Point 4:
- Figure 1: although some of the described drugs with their therapeutic target are displayed, the figure does not really add any insight how they mechanistically interact with the pathogenesis of the PTCL and should therefore be revised.
Response 4: The figure has been revised.
Point 5:
- paragraph 3.3: add "Lobello, C., Tichy, B., Bystry, V. et al. STAT3 and TP53 mutations associate with poor prognosis in anaplastic large cell lymphoma. Leukemia 35, 1500–1505 (2021). https://doi.org/10.1038/s41375-020-01093-1" reference
Response 5: Change has been made on lines 230-231.
Point 6:
- line 280: sometimes PD-1 is reported as protein sometime as gene pcdc1, it has to be consistent
Response 6: Could not find any mention of pcdc1.
Point 7:
- abbreviation in the entire manuscript need to be controlled
Response 7: Fixed the abbreviation for brentuximab vedotin.
Point 8:
- line 349: ref needs to be added as well as ref. to table2
Response 8: Reference 66 was already in text. It has been added to Table 2.
Point 9:
- paragraph 6.8: FDA crizotinib approval for ALCL have to be mentioned here
Response 9: Change has been made.
Reviewer 2 Report
The authors have prepared a useful and comprehensive review of current knowledge regarding PTCL biology. This is an excellent reference for clinicians who are seeking further information on mutations identified on genetic profiling reports. Figure 1 and visual depiction of therapeutic targets is particularly useful.
Author Response
Reviewer 2 had no comments/suggestions
Reviewer 3 Report
The authors review the current state of the knowledge of T-cell lymphoma pathogenesis and current treatment option. Overall the manuscript provides satisfactory information on the subject. However, I have some points regarding the way the manuscript is written:
- The statistics or info that are cited in the text are in some moments not supported by the reference. E.g. in the Introduction section: „PTCLs constitute 15-20% of aggressive non-Hodgkin lymphomas (NHLs) and 5-10% 37 of all NHLs” – could you provide reference for confirming this statistics, as well as check and refill where necessary.
- Please, try to simplify Table 1. It’s a bit chaotic. E.g. you the reader does not know if rearrangements of CD28 could be classified as RHOA or TCR pathway. I recommend to divide the columns so that the disrupted molecular pathway could go with disrupted genes in the 3rd column.
- In paragraph 3.1 please explain first what Tfh means. Also please, give the full name for the abbreviated names of genes and proteins when possible.
- Paragraph 3.1 in my opinion could be re-writed. The paragraph just lists the proteins disrupted and therefore seems not attractive to be read. Please, try to unify it to be more reading-friendly. I would strongly recommend for the authors to have the manuscript checked by the native speaker f so that they could improve the manuscript style.
- Line 350: please, explain the BV abbreviation
Author Response
Response to Reviewer 3 Comments
Point 1: The statistics or info that are cited in the text are in some moments not supported by the reference. E.g. in the Introduction section: „PTCLs constitute 15-20% of aggressive non-Hodgkin lymphomas (NHLs) and 5-10% 37 of all NHLs” – could you provide reference for confirming this statistics, as well as check and refill where necessary.
Response 1: Reference for the mentioned sentence has been added.
References have also been added for sentences on lines 126, 135-138.
Point 2: Please, try to simplify Table 1. It’s a bit chaotic. E.g. you the reader does not know if rearrangements of CD28 could be classified as RHOA or TCR pathway. I recommend to divide the columns so that the disrupted molecular pathway could go with disrupted genes in the 3rd column.
Response 2: Table 1 has been graphically revised.
Point 3: In paragraph 3.1 please explain first what Tfh means. Also please, give the full name for the abbreviated names of genes and proteins when possible.
Response 3: Tfh has already been mentioned on line 66.
Point 4: Paragraph 3.1 in my opinion could be re-writed. The paragraph just lists the proteins disrupted and therefore seems not attractive to be read. Please, try to unify it to be more reading-friendly. I would strongly recommend for the authors to have the manuscript checked by the native speaker f so that they could improve the manuscript style.
Response 4: Minor changes were made.
Point 5: Line 350: please, explain the BV abbreviation
Response 5: Change made.